**⊘ | Open Peer Review** | Genetics and Molecular Biology | Research Article

# Probing the function of *Streptomyces albidoflavus* J1074 gene *XNR_5296* for SPOUT family ribose methyltransferase

Vasylyna-Marta Tseduliak,[1] Oksana Koshla,[1] Sophie N. Mulartschyk,[2] Virginie Marchand,[3] Yuri Motorin,[3,4] Mark Helm,[2] Bohdan Ostash[1]

**ABSTRACT**  *Streptomyces albus* (*albidoflavus*) J1074 is one of the preferred streptomycete chassis strains for the expression of specialized metabolite biosynthetic gene clusters. Leucyl tRNA gene *bldA* is one of the regulatory switches that, through delayed translation of its cognate codon UUA, confines the production of specialized metabolites to a stationary phase. An integral step in the maturation of the tRNA$_{UAA}$ is its post-transcriptional tRNA modifications (PTTMs), which are poorly understood. Exploring the installation of BldA PTTMs may reveal their cross-talk with antibiotic biosynthesis regulatory pathways and offer new ways to manipulate specialized metabolism in *Streptomyces*. In this work, we focused on the J1074 gene *XNR_5296*, coding for a SPOUT family tRNA methyltransferase homologous to *Escherichia coli* TrmL that methylates the ribose residue of uridine (2′-O-methyluridine or Um) at the wobble position of leucyl tRNA$_{UAA}$. First, we revisited the diversity of modified nucleosides for the wild-type strain and suggest that wobble uridine in tRNA$^{Leu}_{UAA}$ is in the form of s$^2$Um. Wobble uridine hypermodifications, such as mnm$^5$s$^2$U (5-methylaminomethyl-2-thiouridine), cmnm$^5$s$^2$U (5-carboxymethylaminomethyl-2-thiouridine), and cmnm$^5$Um (5-carboxymethylaminomethyl-2′-O-methyluridine), found in enterobacteria, could not be confirmed for J1074. Second, while an *XNR_5296* knockout did not diminish the formation of s$^2$Um, it did lead to a strong decrease in the abundance of Um in total nucleoside hydrolyzates. The loss of Um32 in leucyl tRNA$_{GAG}$, as well as the loss of 2′-O-methylated cytosine 32 (Cm32) in prolyl tRNA$_{GGG}$, was confirmed by RiboMethSeq profiling of the mutant. Our results are reminiscent of the abrogated TrmJ function responsible for position 32 C/U methylation in Gram-negative bacteria. Notably, our findings are the first demonstration of TrmJ-controlled methylation in Gram-positive bacteria. This work expands the understanding of tRNA modification systems in streptomycetes and their potential impact on specialized metabolite production.

**IMPORTANCE** Post-transcriptional modifications are ubiquitous in tRNAs, where they play important structural and regulatory roles. As the types of modified nucleosides and their genetic control differ even between closely related bacterial taxa, there is a need to study them across the entire phylogenetic tree. We recently initiated studies of genetics and chemistry of tRNA modifications in streptomycetes, one of the most prolific producers of specialized metabolites of immense practical value (antibiotics, anticancer drugs, to name just a few). A point of special interest was the modifications of leucyl tRNA$_{UAA}$, the only one capable of decoding the rarest in *Streptomyces* codon UUA. In a search for a TrmL homologue responsible for 2′-O-methylation of the wobble nucleoside 34 (U) ribose of tRNA$_{UAA}$, we probed the function of gene *XNR_5296*. *XNR_5296* knockout led to the loss of 2′-O-methylated uridine 32 (Um) in leucyl tRNA$_{GAG}$ and 2′-O-methylated cytosine 32 (Cm) in prolyl tRNA$_{GGG}$. This result, as well as *in silico* analysis, suggests parallels between Xnr_5296 and the *Escherichia coli* TrmJ enzyme responsible for U/C methylation at position 32 of glutaminyl tRNA$_{UUG}$

**Peer Reviewer** Rebecca Alexander, Wake Forest University, Winston-Salem, North Carolina, USA

Address correspondence to Bohdan Ostash, bohdan.ostash@lnu.edu.ua.

Vasylyna-Marta Tseduliak and Oksana Koshla contributed equally to this article. The author order was determined based on increasing seniority.

The authors declare no conflict of interest.

See the funding table on p. 11.

and tRNA$_{CUG}$, methionyl tRNA$_{CAU}$, seryl tRNA$_{UGA}$, and tryptophanyl tRNA$_{CCA}$, although the *Streptomyces* counterpart methylates different tRNA species. Thus, our work reveals previously unreported tRNA modification and its gene in *Streptomyces* and serves as a stepping stone to further interrogate the functions of highly paralogous SPOUT family methyltransferases in this important bacterial genus.

**KEYWORDS** *Streptomyces albidoflavus* (*albus*) J1074, genes, bioactive natural products, regulation, tRNA, post-transcriptional tRNA modifications, methyltransferases

The actinomycete genus *Streptomyces* includes some of the most dexterous natural chemists in the microbial world (1), whose specialized metabolites have been converted into valuable products, of which antibiotics are perhaps the most prominent. The specialized metabolism of *Streptomyces* is orchestrated by a large network of regulatory genes that define the onset and level of production of a small molecule (2). A detailed knowledge of the regulatory network will facilitate the rational construction of antibiotic overproducers and the discovery of novel bioactive compounds (3).

While much of the regulation of specialized metabolism in streptomycetes happens at the transcriptional level (2), there are also mechanisms impacting the stage of translation. The regulatory switch based on gene *bldA* for leucyl tRNA (tRNA$^{Leu}_{UAA}$) is perhaps the most studied type of such a mechanism in *Streptomyces*. In their GC-rich genomes, UUA codon is the rarest one, and its decoding appears to be strictly dependent on a single cognate tRNA. There is evidence for temporal control of accumulation of mature *bldA* tRNA (4), which leads to a gap between occurrence of UUA-containing transcript and its translation (5–7). The principal genes through which the Bld-phenotype is manifested are TTA-containing pleiotropic regulatory gene *adpA* (*bldH*) needed for normal morphological development (8), and TTA-containing pathway-specific regulatory genes controlling the expression of the antibiotic biosynthesis genes (5, 9).

The level of charged *bldA* tRNA is thought to determine the rate of translation of UUA-containing transcripts, effectively serving as a bottleneck in the overall process of expression of TTA-containing gene. However, it is not known what, in the first place, determines the delayed occurrence of charged tRNA$^{Leu}_{UAA}$. The exact molecular mechanisms behind the final phenotype of *bldA* mutants are complex and influenced by the environment. Particularly, on certain solid media, *bldA* mutants still produce aerial hyphae and spores (10). Overexpression of a TTA-containing regulatory gene for biosynthesis of actinorhodin restored the production of the latter in *S. coelicolor* (11). Also, mistranslation of UUA-containing transcripts in *bldA* mutants was reported (12). Although suspected initially, in *S. coelicolor,* there are no temporal differences in transcription of tRNA genes, including the one for tRNA$^{Leu}_{UAA}$ (13).

Nucleosides in tRNA are extensively modified to ensure proper tRNA folding, stability, and decoding. Probably the most important part of the tRNA to be modified is the anticodon loop. Positions 34 and 37 have the most diverse set of post-transcriptional tRNA modifications (PTTMs), ranging from 2′-O-ribose methylation to complex hypermodifications (14–16). Positions 32 and 34 of tRNAs with UXX anticodon may share the same PTTM, 2′-O-methyluridine (Um). Some PTTMs mentioned in this work are depicted in Fig. 1.

For *Escherichia coli* (and a few other well-studied models), PTTMs are known to respond to environmental conditions and influence tRNA function (14–16, 23–25). We suggested that PTTMs may play a similar role in *bldA*-mediated control of specialized metabolism of *Streptomyces* (26) and went on to reveal Bld-like phenotypes (21, 27) for mutants deficient in prenylation and thiomethylation of adenosine in position 37 (ms$^2$io$^6$A37). Another recent work highlighted the importance of methylated tRNA (m$^1$A in position 58) for the specialized metabolism of *Streptomyces venezuelae* (28). Besides these reports, nothing is known about genetic identity and biological significance of many other streptomycete PTTMs (29).

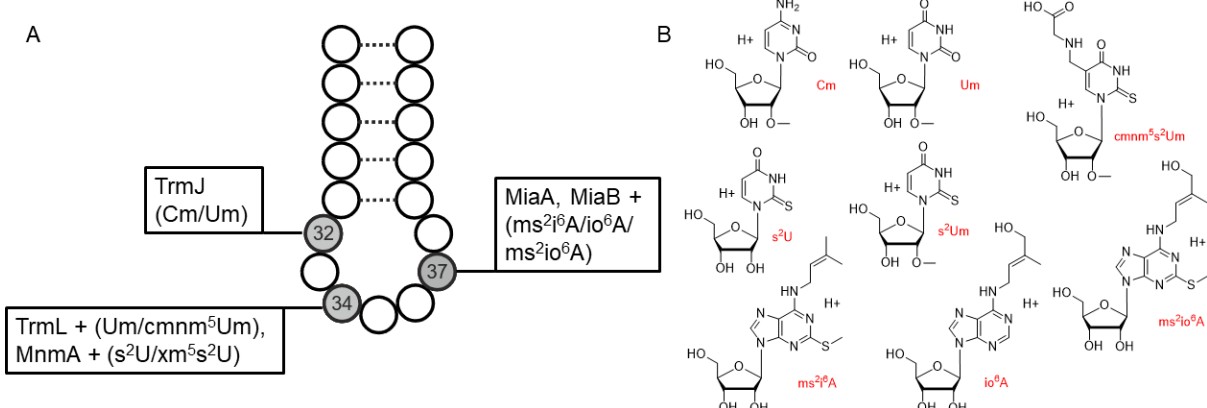

**FIG 1** Some known anticodon loop PTTMs relevant to this work, their biosynthetic enzymes and structures. (A) Selected PTTMs and the corresponding enzymes for modifying the UXX anticodon loop. Position 32 can harbor ribose 2′-O-methylation in *Escherichia coli*, but not in *Bacillus subtilis* (17, 18). Wobble uridine 34 can be methylated, thiolated, or hypermodified in *E. coli* and *B. subtilis* (18, 19). Position 37 is represented by 2-methylthio-N6-isopentenyladenosine (ms$^2$i$^6$A) in both bacteria (18, 20), and by 2-methylthio-N6-(cis-hydroxyisopentenyl) adenosine (ms$^2$io$^6$A) or N6-(cis-hydroxyisopentenyl)adenosine (io$^6$A) in *Salmonella typhimurium* and *S. albidoflavus* (21, 22). The "+" sign indicates additional enzymes required for nucleoside hypermodification. (B) Some PTTMs characteristic of the UXX anticodon loop. The position of the 2-thio-2′-O-methyluridine (s$^2$Um) modification is not yet known, but we tentatively assume its presence in a wobble position in *S. albidoflavus* (see explanation below).

Here, we set out to probe the functions of putative tRNA methyltransferase (MT) genes in this genus, given the fundamental importance of methylation for decoding and translational fidelity (15, 23). The SPOUT family of MTs in our model species, *Streptomyces albidoflavus* J1074, appeared especially intriguing, as it likely harbors the counterpart of *E. coli* TrmL for methylation of the ribose residue of tRNA$^{Leu}_{UAA}$ wobble uridine, presumably to yield Um or congeners such as 2-thio-2′-O-methyluridine (s$^2$Um) (19). The presence of several paralogs within the family and their distant similarity to known MTs implies that experimental verification of any TrmL candidate has to be carried out. To this end, we here present an updated view of PTTMs present in J1074 tRNA hydrolyzates as well as a more detailed bioinformatic scrutiny of Xnr_5296 sequence and structure, as the latter was considered the most plausible TrmL candidate due to phylogenetic considerations (29). Notably, Xnr_5296 domain organization is different from that of TmJ; the tertiary structure of Xnr_5296 was no more similar to TrmL than it was to TrmH or TrmJ. Knockout of *XNR_5296* did not abrogate the accumulation of s$^2$Um. The latter is a presumed precursor to hypermodified wobble uridines, such as 5-methylaminomethyl-2-thio-2′-O-methyluridine (mnm$^5$s$^2$Um) and 5-carboxymethyla-minomethyl-2-thio-2′-O-methyluridine (cmnm$^5$s$^2$Um); as a stand-alone modification, it has also been discovered in archaea and actinomycete *Mycobacterium bovis* (30–32). Rather, we observed a drop of Um 32 abundance in tRNA$^{Leu}_{GAG}$ and 2′-O-methylated cytosine 32 (Cm) in tRNA$^{Pro}_{GGG}$, which is an indicator of impaired TrmJ function. The latter MT is responsible for position 32 of the anticodon stem Cm or Um formation in *E. coli* tRNA$^{Gln}_{UUG}$, tRNA$^{Gln}_{CUG}$, tRNA$^{Met}_{CAU}$, tRNA$^{Ser}_{UGA}$, and tRNA$^{Trp}_{CCA}$ (17, 33). Taking into account these findings as well as the absence of (c)mnm$^5$U pathway in J1074 (29), we propose tentatively that the wobble uridine of *bldA* tRNA is in the form of s$^2$Um. The implications of our findings are discussed within the context of *bldA*-mediated regulation as well as a wider context of biology of actinomycete PTTMs.

## RESULTS

### An updated description of modified nucleosides produced by *Streptomyces albidoflavus* J1074

Our recent description of the chemical diversity of J1074 RNA nucleoside modifications, while revealing those common to most of the bacteria (29), remained inconclusive with

regard to several PTTMs important in the context of *bldA* tRNA of *Streptomyces*. We failed to reveal J1074 homologs of the verified bacterial Mnm proteins involved in hypermodified wobble uridine (c)mnm$^5$s$^2$Um, and the modification itself was present below the reliable detection limit of the MS machine. Instead, we observed s$^2$Um, a presumed intermediate toward the aforementioned PTTM, and also a likely terminal PTTM for some bacteria (see the Introduction). Its presence in the well-studied models, such as *E. coli* or *Bacillus*, was not reported. We approached this issue on the basis of a more careful isolation and preparation of tRNA hydrolyzates, as detailed in Materials and Methods.

Our current findings mostly go in line with the previous results. Twenty PTTMs out of 26 that we reported in 2023 were re-confirmed in this work (Table S1), including s$^2$Um. The abundance of the latter was high and reproducible enough to consider it a reliably detected PTTM (Fig. S1). The m$^5$U and m$^2$A modifications were observed at the MS detection limit, likely because the replete medium tryptic soy broth (TSB) does not support their abundant production (an obstacle we experienced previously as well [29]). We also did not confirm the presence of (c)mnm$^5$s$^2$Um or precursors thereof, which is in line with the reported absence (29) in J1074 of homologues of *E. coli* MnmEG family proteins. Taking into account these results and previous *in silico* efforts, we assume tentatively that (c)mnm$^5$s$^2$Um is not present in J1074; s$^2$Um is the most likely candidate for wobble uridine modification in tRNA having anticodon UXX (see the Discussion).

At the same time, we observed two new nucleoside modifications for *S. albidoflavus*: 2-lysidine (k$^2$C) and 4-acetylcytidine (ac$^4$C) (see Table S1 and Fig. S2 and S3 for LC-MS/MS data). The detection of k$^2$C is in line with the presence within the *S. albidoflavus* genome of gene *XNR_3428* (29) for the apparent ortholog of k$^2$C biosynthetic enzyme TilS (tRNA(Ile)-lysidine synthetase). An acetyltransferase responsible for ac$^4$C biosynthesis in *S. albidoflavus* has yet to be identified. The MS data on newly identified modifications appear plausible, but further investigation is needed.

## Structural bioinformatics attempts to understand Xnr_5296 function

J1074 genome encodes 89 SAM-dependent MTs (29), of which at least 22 are related to DNA/RNA modification, according to their original annotation. None of them was experimentally studied. Here, we focused on an MT Xnr_5296, previously considered the most plausible candidate for methylation of *bldA* wobble uridine position 34 (Um34), by analogy to *E. coli* TrmL (29). According to a Conserved Domain Database (CDD) search, Xnr_5296 contains a SpoU-superfamily domain (E value 7.36e$^{-74}$), which belongs to the SPOUT MT class (34). Furthermore, the CDD revealed that the SpoU domain of Xnr_5296 belongs to the COG0566, TrmH family, which includes tRNA G18 ribose-2′-O-MTs. SPOUT MTs' in-depth bioinformatic analysis (35) suggested clustering of COG0566 (TrmH family), COG0565 (TrmJ family), and COG0219 (TrmL family) and their monophyletic origin, with multiple duplication events in the cluster evolution. The UniProt search reveals an eL30 protein-like N-terminal domain (NTD) and the SPOUT domain at the C-terminus (CTD) (Fig. 2A). The SPOUT is a diverse class of MTs involved in the methylation of ribose, bases, and, rarely, proteins. In *E. coli*, neither TrmJ (Um/Cm32) nor TrmL (Um/Cm34) possesses an NTD as found in Xnr_5296, while TrmH MTs (Gm) have one. Multiple sequence alignment of Xnr_5296 with experimentally validated TrmJ and TrmL proteins further showed that Xnr_5296 features a unique NTD not found in any of the aligned sequences ( Fig. S4 and S5).

The combination of a topological trefoil knot in the catalytic domain and the absence of a conserved catalytic core is a hallmark of SPOUT MTs (35, 36). As shown in Fig. 2B, Xnr_5296 retains most of the catalytic and SAM-binding residues essential for TrmJ function (denoted by black and red arrows, respectively). However, unlike canonical bacterial TrmJ enzymes, Xnr_5296 lacks the TrmJ-specific "TXARXR" motif, which is critical for binding both SAM and tRNA (33), as illustrated in the expanded sequence alignment (ESM Fig. S4). Given the absence of this feature in Xnr_5296, we believe that the NTD of Xnr_5296 is involved in substrate recognition and tRNA binding.

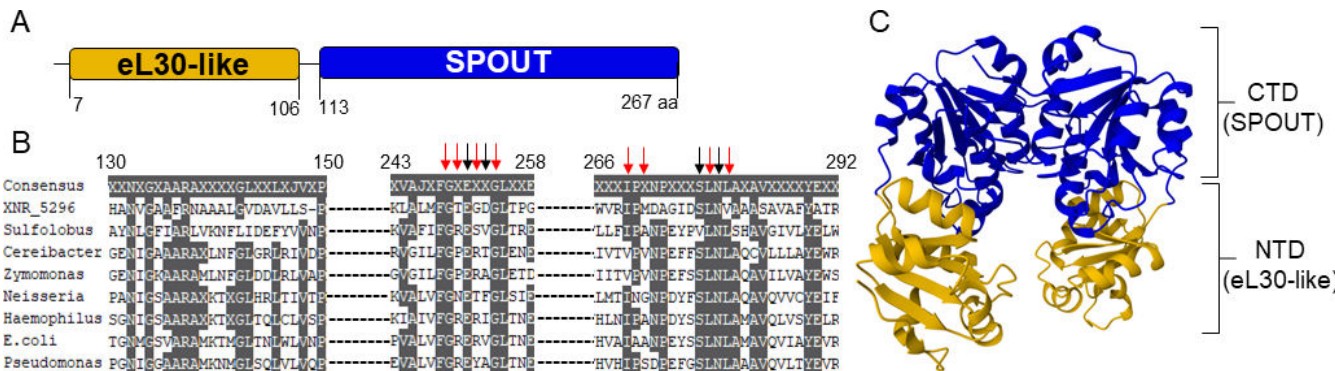

**FIG 2** Bioinformatic analysis of *XNR_5296* gene product. (A) Putative domain structure of Xnr_5296 based on UniProt peptide search. NTD is represented by ribosomal protein eL30-like superfamily domain, and CTD is SPOUT-type methyltransferase domain. (B) MSA was performed using TrmJ sequences from different organisms that have been verified experimentally (the sequence accession numbers used are listed in the Materials and Methods). Black arrows indicate the typical catalytic residues, and red arrows indicate the typical SAM-binding residues (according to reference 35). (C) AlphaFold predicted structure of Xnr_5296. The color code for domain indications is the same as in panel A.

We modeled the tertiary structure of Xnr_5296 using AlphaFold and compared the model to structurally characterized MTs available in the PDB. The predicted Xnr_5296 contains two distinct domains (Fig. 2C), unlike the other tRNA 2′-O-ribose MTs. Superposition of Xnr_5296 and TrmJ, TrmL, and other MTs from the PDB did not reveal any highly similar tertiary organization of the proteins (ESM Fig. S6). The only exception was the Xnr_5296–AviRb pair. The latter is an MT encoded within the avilamycin biosynthetic gene cluster of *Streptomyces viridochromogenes*. AviRb methylates ribose at U2479 in the 23S rRNA, protecting the bacterium from the toxic action of avilamycin (37). Nevertheless, Xnr_5296 is not an ortholog of AviRb; according to reciprocal best BLASTP searches, the most likely candidate for that role for *S. albidoflavus* J1074 is Xnr_4256. Hence, neither domain nor tertiary structure analysis of Xnr_5296 provides a reliable indicator of its function.

## *XNR_5296* knockout points to its involvement in U modification beyond wobble position

We replaced the gene *XNR_5296* with a hygromycin resistance cassette and verified, via PCR, the resulting knockout strain Δ5296 ( Fig. S6). As compared to the parent J1074 strain, Δ5296 showed no significant morphological or antibiotic biosynthesis defects (Fig. 3), the usual companions of impaired *bldA* function (10, 21). Nevertheless, J1074 and Δ5296 differed in the types of methylated uridines they produce (Fig. 4). While the nucleoside hydrolyzates from the parental strain harbored both Um and s$^2$Um, Δ5296 still produced s$^2$Um but had a dramatically decreased amount of Um. No other changes in the PTTM patterns have been observed. Notably, both J1074 and Δ5296 accumulated significant quantities of two previously undescribed nucleoside fractions (*m/z* 258 Da) eluting at 8 and 10 min, with their protonated forms (*m/z* 259 Da) appearing as two major peaks in Fig. 4.

## The *XNR_5296* knockout perturbs methylation of position 32 in tRNA$^{Leu}_{GAG}$ and tRNA$^{Pro}_{GGG}$

To identify tRNA substrates of *XNR_5296*-encoded protein, we performed RiboMethSeq analysis (see Materials and Methods) of tRNA 2′-O-methylation in J1074 and Δ5296 strains. Inspection of protection profiles revealed substantial differences between the two strains for tRNA$^{Leu}_{GAG}$ and tRNA$^{Pro}_{GGG}$, where protection of U32 and C32, respectively, was strongly affected (Fig. 5). Cleavage profiles for all other inspected tRNAs were identical, and no change was detected upon *XNR_5296* gene deletion. These results

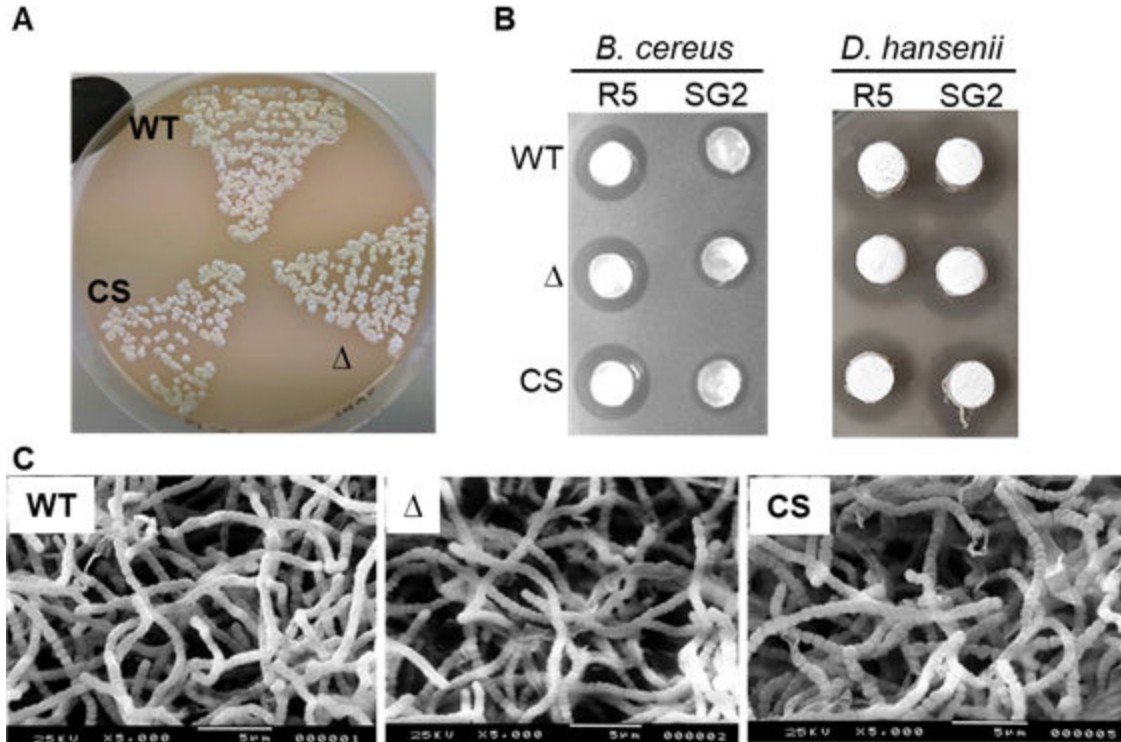

**FIG 3** The *XNR_5296* mutant did not differ from the parental strain in its morphogenesis and antibiotic activity on solid media. The cultures of the parental strain (WT; harbors empty vector pTES), *XNR_5296* null mutant (Δ), and its complementation strain (CS) grown on soy-flour mannitol (SFM) agar for 120 h (A) have similar appearance. Agar plugs cut off the lawns grown on R5 and SG2 agars (120 h) showed that the strains display similar activity against *Bacillus cereus* and *Debaryomyces hansenii* (B). Scanning electron microscopy images (C) of the SFM agar-grown cultures shown in part A.

allow assigning *XNR_5296* gene activity to 2′-O-methylation of Um/Cm32 in *S. albidoflavus* tRNA$^{Leu}_{GAG}$ and tRNA$^{Pro}_{GGG}$.

We cannot help noting a few more peculiarities of RiboMethSeq data for the wild-type strain (which can be found at the ENA archive of the raw data, PRJEB98478). First, protected Um34 residues (or derivatives) were detected in tRNA$^{Gln}_{UUG}$, tRNA$^{Glu}_{UUC}$, and tRNA$^{Lys}_{UUU}$. Second, numerous protected residues were also observed at position 12 in many tRNAs. This tRNA position was never reported to be 2′-O-methylated. Finally,

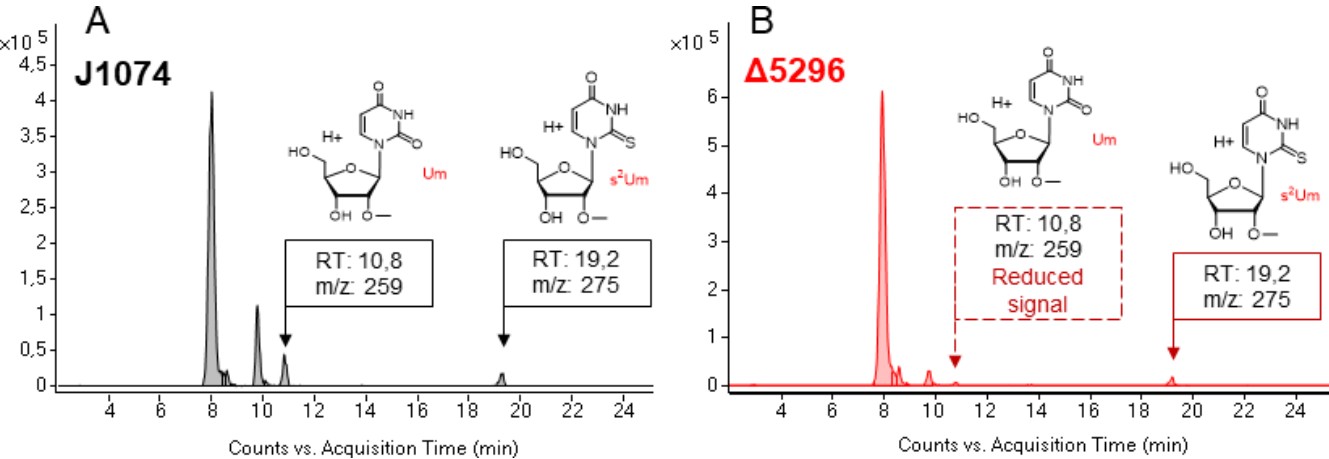

**FIG 4** Knockout of *XNR_5296* diminishes the Um accumulation, but does not affect s$^2$Um production. (A) Composite extracted ion chromatogram (CEIC) showing the Um and s$^2$Um traces in total tRNA hydrolyzate from the wild-type strain (J1074). (B) CEIC from the mutant strain (Δ5296) revealed the presence of s$^2$Um and the absence of Um.

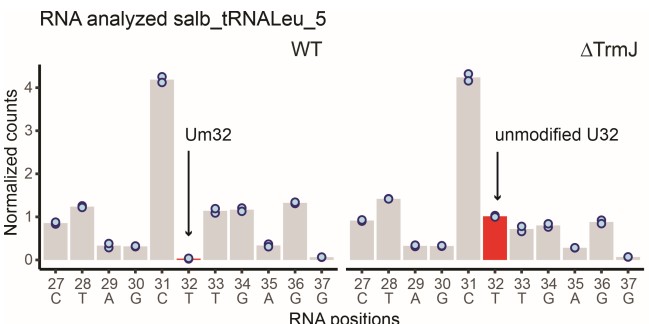
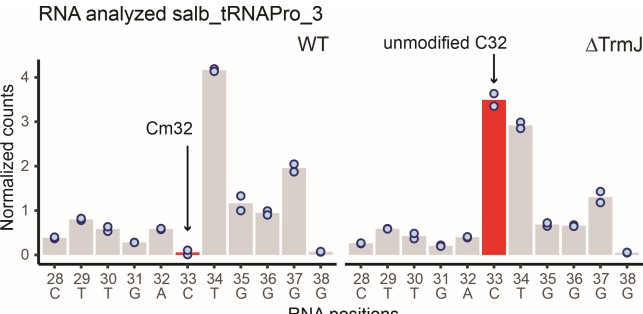

**FIG 5** RiboMethSeq protection profiles obtained for *S. albidoflavus* tRNA$^{Leu}_{GAG}$ and tRNA$^{Pro}_{GGG}$ from WT(J1074) and Δ5296 strains. Cleavage intensity is shown as normalized cleavage value, representing reads' count number divided by the average for the region. The experiment was performed in biological duplicate, and individual points are shown on the graphs. The region for ±5 nt positions is shown, sequence and sequential numbering of nucleotides are given at the bottom. Due to the presence of variable nucleotides in tRNA D-loop, the sequential numbering of nucleotides may not correspond to tRNA numbering convention. Modified and unmodified Um/Cm residues are shown by arrows and colored in red. The identity of the strains is shown on the top.

no protected Gm18 was observed (confirming the absence of canonical TrmH activity, as expected for Gram-positive bacteria), although Gm is clearly present in J1074 tRNA hydrolyzates (this work and 21).

## DISCUSSION

RNA methylations constitute the most abundant type of PTTM and are critical (among other roles) for proper biogenesis of tRNAs and fidelity of codon-anticodon recognition (17, 19, 23, 28, 32, 38). They are installed onto RNAs by highly similar MTs sharing an intricate evolutionary history. This is especially true for the SPOUT family of RNA MTs (35, 36, 39). While in model microorganisms the focus has shifted to elucidating the molecular details of SPOUT MT action (40), for many bacterial classes, the genes responsible for the most conserved PTTMs remain unknown. We encountered this kind of situation while studying how leucyl tRNA$_{UAA}$, or *bldA*, could control the onset of antibiotic production by *Streptomyces* (26). While some of the PTTMs have been elucidated since then in the model species *S. albidoflavus* J1074 (21), there are no experimental data on its RNA MTs. Here, we focused on one gene, *XNR_5296*, encoding a homolog of *E. coli* SPOUT family MTs, such as TrmL, TrmH, and TrmJ. We reproducibly detect two known ribose-methylated species of uridine in J1074, Um and s$^2$Um. We further show that *XNR_5296* knockout does not affect J1074 ability to sporulate and produce antibiotics, at least under typical laboratory conditions. The knockout strongly decreases Um accumulation, but not Cm, Gm, or s$^2$Um. *XNR_5296* knockout led us to conclude that this gene controls the biosynthesis of Um, and methyl groups within Um and s$^2$Um emanate from the activity of distinct MTs. We next resorted to the RiboMeth-Seq approach (41) to determine the exact positions of *XNR_5296*-controlled methylations and showed that the absence of the aforementioned gene led to loss of methylation at position 32 of tRNA$^{Leu}_{GAG}$ (U32) and tRNA$^{Pro}_{GGG}$ (C32). Our inability to pinpoint the effect of *XNR_5296* knockout on the abundance of Cm might stem from the relatively small contribution of tRNA$^{Pro}_{GGG}$ to the overall pool of Cm. Hence, in its activity, Xnr_5296 MT resembles TrmJ of *E. coli* that produces Um/Cm32 in a number of tRNAs, albeit not the ones revealed in our work. We note that Xnr_5296 is clearly different from canonical TrmJ of *E. coli*. Xnr_5296 has an additional N-terminal domain (the one seen in TrmH MT involved in Gm production), its sequence lacks one of the TrmJ hallmark motifs; and TrmJ MTs so far were described only for enterobacteria.

The PTTM s$^2$Um, discovered for the first time over 30 years ago in *Mycobacterium*, was believed to be a precursor for more complex nucleoside modifications (30). In J1074, we could not detect U34 hypermodifications known for the *E. coli* UAA anticodon (e.g., (c)mnmUm, etc.) despite repeated efforts in this work and in the past. Furthermore, there are no gene candidates in J1074 for MnmEG enzymes complex to produce such

a modification (29). 2-thiouridine in bacteria so far has been found only in the wobble position of several anticodons, yet not in UAA (42). On the other hand, 2-thiolated wobble uridines are not known to carry methylated ribose (43). Thus, we observe a novel combination of known modifications of a uridine residue and suggest that $s^2Um$ is present in the wobble uridine position of the UAA anticodon. Identification and manipulation of genes behind this PTTM will help confirm or reject this hypothesis. All said above underscores our belief that actinomycetes are a source of unusual nucleoside modifications. Their comprehensive investigation will definitely broaden our knowledge of the roles of tRNA as regulators of valuable biological processes.

## MATERIALS AND METHODS

### Plasmids, microorganisms, and culture conditions

Plasmids and strains used in this work are described in Table S2 (ESM). *Streptomyces albidoflavus (albus)* SAM2, a derivative of J1074 carrying a single *attB*$^{\varphi C31}$ integration site (44), was used to construct the Δ5296 mutant. *E. coli* was grown at 37°C (45) except for BW25113 (pKD46), which was incubated at 30°C prior to the λ-Red recombination event (46). For intergeneric matings and phenotypic examination, *S. albidoflavus* strains were grown on soy-flour mannitol (SFM) agar (47). Streptomycetes and bioassays were incubated at 30°C, unless otherwise stated. For total tRNA extraction, *S. albidoflavus* strains were grown in TSB (Merck Millipore, cat no 1.05459) for 48 h. To reveal endogenous antibacterial and antifungal activities, *S. albidoflavus* strains were grown on R5 and SG2 agar (29). Where needed, strains were maintained in the presence of apramycin (25 µg/mL) or hygromycin (100 µg/mL); chromogenic substrate X-Gal and inducer IPTG were added to the media to a final concentration of 50 and 20 µg/mL, respectively.

### Plasmid construction

The *XNR_5296* knockout plasmid was constructed using the suicide vector pKC1132 (48). The gene was amplified with approximately 1 kb flanks from SAM2 chromosome using primers xnr5296_dup and xnr5296_drp (3 kb product). A list of all the primer sequences used in this study is provided in Table S3 (ESM). The digestion of the amplicon with HindIII and EcoRI restriction endonucleases and its subsequent cloning into the HindIII-EcoRI sites of pKC1132 resulted in the production of pKC5296KO plasmid. Hygromycin resistance cassette *hyg* was amplified from patt-shyg (49) with primers trmL_red_up and trmL_red_rp (1.4 kb product). The knockout plasmid, pKC5296KOhyg, was generated by replacing the *XNR_5296* gene in pKC5296KO with *hyg*, using the *E. coli* recombineering strain BW25133 (pKD46). Recombinant plasmids were verified by PCR and sequencing. To construct plasmid pTES5296, gene *XNR_5296* was amplified with xnr5296_Xbalup and xnr5296_EcoRIrp (810 bp product) from SAM2 chromosome, digested with XbaI and EcoRI restriction endonucleases, and cloned downstream of *ermEp* into pTES (42).

### Generation and verification of the *S. albidoflavus* recombinant strains

For knockout, pKC5296Kohyg plasmid was used, and for complementation experiment, pTES5296 plasmid was employed to complement *XNR_5296* loss in the mutant. All constructs were transferred into *S. albidoflavus* conjugally from *E. coli* ET12567 (pUZ8002), as described elsewhere. The *XNR_5296* knockout strain of *S. albidoflavus* was selected for hygromycin resistance (gene replacement with *hyg*) and apramycin sensitivity (loss of vector sequences). PCR was employed to confirm the presence of the plasmids and expected gene replacements in *S. albidoflavus*, as detailed in ESM Fig. S7.

### Purification of total tRNA from *S. albidoflavus*

The *S. albidoflavus* cultures for total tRNA isolation were cultivated as described in the previous study (29) with minor modifications as described below. *S. albidoflavus* strains

were grown in 250 mL Erlenmeyer baffled flasks with 25 mL of TSB, and spore suspension of each strain was used for inoculation. After growing the strains for 24 h in an incubator shaker (120 rpm, 30°C), 10 mL of the precultures was used to inoculate 500 mL baffled flasks with 100 mL of TSB. Biomass was harvested after 48 h of cultivation (120 rpm, 30°C).

The general procedure for the guanidinium thiocyanate total tRNA purification method was carried out as indicated in our previous work (29). The concentration of tRNA in the samples was quantified using NanoDrop, and the purity of the tRNA was checked by capillary electrophoresis on the Agilent TAPE station. If traces of rRNA were detected, total tRNA was separated from rRNA by further purification via urea PAGE gel electrophoresis. For gel preparation, 50 mL of 10% urea PAGE premix solution (40% Rotiphorese sequencing gel-concentrate, 50% Rotiphorese sequencing gel-buffer concentrate solution, 10% Rotiphorese sequencing gel-buffer concentrate) was mixed with 200 µL of APS (10% ammonium persulfate solution, APS ≥ 98%, practical grade p.a., Carl Roth) prior to the addition of 20 µL of TEMED (*N,N,N´N´*-tetramethylethylenedi-amine, ≥99%, p.a., Carl Roth). The gel solution was immediately poured between two glass plates (20 cm × 20 cm × 0.01 cm), and a suitable comb was inserted to form the loading pockets. After removing the comb, the polymerized 10% denaturing PAGE gels were pre-run for 10 min at 12 W in 1× TBE buffer. A total of 40 µg of the sample mixed with 2× formamide gel-loading buffer (1:1 mixture of sample with 90% vol/vol formamide in 1× TBE) was loaded into each pocket. The gels were run for 90 min at 12 W. To visualize nucleic acids after electrophoresis, the gels were stained in 100 mL 1× GelRed solution for 15 min and scanned on the Typhoon TRIO+ Variable Mode Imager. The excitation light wavelength was set at 532 nm and the emission light wavelength at 610 nm. Bands containing tRNA were excised from the acrylamide gel, cut into small pieces, and crushed. The gel pieces were frozen at −80°C for 5 min before 200 µL of 0.5 M $NH_4OAc$ was added. The samples were incubated overnight at 650 rpm and 15°C to allow tRNA diffusion out of the gel. To separate the tRNA from the gel, the samples were filtered through Nanosep centrifugal filters, while the remaining gel pieces were washed once with 50 µL 0.5 M $NH_4OAc$. The isolated tRNA in the flow-through was ethanol precipitated as described above. The tRNA concentration was determined by NanoDrop device (UV-VIS spectrophotometer) with a wavelength of 260 nm.

## tRNA digestion and liquid chromatography-tandem mass spectrometry (LC-MS)

For qualitative LC-MS analysis, up to 5 µg of total tRNA per replicate for each sample (SAM2, *XNR_5296* mutant, and *E. coli* $^{12}$C RNA as reference and internal standard) was digested to nucleoside level using 0.6 U nuclease P1 from *Penicillium citrinum* (Sigma-Aldrich), 0.2 U snake venom phosphodiesterase from *Crotalus adamanteus* (Worthington), 0.2 U bovine intestine phosphatase (Sigma-Aldrich), 10 U benzonase (Sigma-Aldrich), and 0.2 µg pentostatin (Sigma-Aldrich) in 5 mM Tris (pH 8) and 1 mM magnesium chloride for 2 h at 37°C in a total volume of 20 µL. The LC-MS measurement was performed on an Agilent 1260 Infinity (II) series HPLC coupled to an Agilent 6470B Triple Quadrupole (QQQ) mass spectrometer with an electrospray ion source (ESI). An amount of 2 µg of digested total tRNA per sample replicate was injected into the LC-MS system and separated on a Synergi Fusion RP-C18 column (250 × 2.0 mm, 4 µM, 80 Å; Phenomenex, Germany) at a temperature of 35°C by using a gradient with a flow rate of 0.35 mL/min. Solvents utilized for the elution consisted of freshly prepared 5 mM ammonium acetate buffer (pH 5.3; solvent A) and LC-MS grade acetonitrile (solvent B; Honeywell). The percentage composition of both solvents was determined by a gradient starting with 100% of solvent A. The proportion of solvent B was increased to 10% at 20 min, to 25% at 30 min, and finally to 80% after 40 min. This state was held for 3 min before initial conditions (100% solvent A) were restored for 14 min. The ESI parameters were defined as follows: 300°C gas temperature, 7 L/min gas flow, 60 psi nebulizer pressure, 400°C sheath gas temperature, 12 L/min sheath gas flow, and 3,000

V capillary voltage. The UV traces of the canonical nucleosides were monitored with a diode array detector at 254 nm. The mass spectrometer was operated in positive ion mode using neutral loss scan (NLS) mode in a mass range of 230–600 Da. The NLS approach was adjusted to monitor the loss of either a ribose moiety (−132 Da) or a 2′-O-methylated ribose moiety (−146 Da), and for pseudouridine dissociation, the loss of two water molecules was detected (−36 Da) (50). In addition, the detection scheme was set to exclude the canonical nucleosides from analysis by omitting the detection of cytidine (*m/z* 244, Rt = 4.6 min), uridine (*m/z* 245, Rt = 6.1 min), guanosine (*m/z* 284, Rt = 9,4 min), and adenosine (*m/z* 268, Rt = 14,1 min) at their respective retention times (50). Data analysis was performed using the Agilent MassHunter Qualitative Analysis software (v10.0). The retention times of different nucleosides were determined from the extracted ion chromatograms for predefined mass transitions of interest and then compared to previous findings (29).

## Phenotypic analysis of *S. albidoflavus*

Lawns of the mutant and parental strains grown on SFM agar for 120 h were used to take photos and for scanning electron microscopy as described in reference 29. Native antibiotic activity of *S. albidoflavus* strains was monitored using agar plug antibiotic diffusion assay. Briefly, strains were grown on SG2 and R5 agar for 5 days. Then agar plugs (Ø 5 mm) were cut off the lawn and stacked on top of TSB agar plates with test culture *D. hansenii* spread immediately prior to the experiments or *B. cereus* spores as it is described above. Halos of growth inhibition around the plugs were observed after 18 h of incubation.

## RiboMethSeq analysis of Nm residues in *S. albidoflavus* tRNAs

Total tRNA (~100 ng) from *S. albidoflavus* cells was subjected to random fragmentation by alkaline hydrolysis in 50 mM sodium-bicarbonate buffer at pH 9.2 and 96°C for 14 min. The reaction was stopped by ethanol precipitation using 3M Na-OAc, pH 5.2, and glycoblue. After centrifugation, the RNA pellet was washed with 80% ethanol and resuspended in nuclease-free water. RNA fragments were end-repaired as previously described (41, 51) and purified using the RNeasy MinElute Cleanup kit according to the manufacturer's recommendations, except that 675 µL of 96% ethanol was used for the RNA binding step. Elution of purified RNA fragments was performed in 19 µL of nuclease-free water. RNA fragments were converted to a library using the NEBNext Small RNA Library Prep Set for Illumina (NEB ref E7330S, USA) following the manufacturer's recommendations. DNA library was quantified using a fluorometer (Qubit 2.0 fluorometer, Invitrogen, USA) and qualified using a High Sensitivity DNA chip on Agilent Bioanalyzer 2100. Libraries were multiplexed and subjected to high-throughput sequencing on an Illumina NextSeq2000 instrument with a 50 bp single-end read mode.

High-quality raw sequencing reads (>Q30) were subjected to trimming using Trimmomatic v0.39 (52) with the following parameters: MINLEN:08, STRINGENCY:7, AVGQUAL:30. Trimmed reads were further processed to keep only short reads with fragmentation-defined 3′-extremities. Selected trimmed reads were aligned to the *S. albidoflavus* tRNA reference sequence using bowtie2 v2.4.4 (53) in end-to-end mode (--no-unal --no-1mm-upfront -D 15 -R 2 -N 0 -L 10 -i S,1,1.15 as other bowtie2 parameters). Only uniquely mapped reads in positive orientation were retained for further analysis. Reads' extremities (5′ and 3′) were counted for each RNA position in the reference, and a cumulative 5′/3′-protection profile was established as described (41, 51), in R/R-studio environment. Specific RiboMethSeq scores (mean, A, B, and C = MethScore [52, 54]) were calculated in ±2 nt interval for all RNA positions and extracted for positions of known Nm residues in the target RNA. If not available, potential Nm positions in RNA sequence were predicted using a medium stringency combination of score mean > 0.99 and score A > 0.55; this combination generally gives the highest stringency in *de novo* Nm detection (55). Quantification of 2′-O-methylated residues was done using MethScore (previously called score C [54] or score C2 [51]). This score is used for

quantification of the modification level since it keeps a linear calibration curve between protection and molar ratio of 2′-methylated residue in RNA.

## Bioinformatic methods

The genome of J1074 (GenBank accession number CP004370.1) was used as a source for primer design and sequence retrieval. PDB and AlphaFold databases were used to retrieve structures of known or predicted SPOUT MTs. Structure inference pipeline AlphaFold2 v2.2.3+49 (56) used to model proteins was accessed via https://console.latch.bio/workflows. MSAs were generated using the Clustal Omega algorithm in Geneious Prime 2025.1.2 software application and modified manually.

## ACKNOWLEDGMENTS

The work was supported by grants BG-19F and M/51-2024 from the Ministry of Education and Science of Ukraine (to B.O.); by the Deutsche Forschungsgemeinschaft (DFG, German Research Foundation), TP C03 in TRR319, Project-ID 439669440 (to M.H.). Research stay of V.-M.T. in Helm lab was supported by FEMS grant (no. 57507437, 2024).

J. Beckwith (Harvard Medical School, USA) is thanked for the gift of *E. coli* BW25113 (pKD46).

## AUTHOR AFFILIATIONS

[1]Department of Genetics and Biotechnology, Ivan Franko National University of Lviv, Lviv, Ukraine
[2]Institute of Pharmaceutical and Biomedical Sciences, Johannes Gutenberg-Universität Mainz, Mainz, Germany
[3]Université de Lorraine, SMP IBSLor, EpiRNASeq Core Facility, Nancy, France
[4]Université de Lorraine, CNRS, UMR7365 IMoPA, Nancy, France

## AUTHOR ORCIDs

Vasylyna-Marta Tseduliak  http://orcid.org/0000-0002-1652-1607
Oksana Koshla  http://orcid.org/0000-0002-2234-1071
Bohdan Ostash  http://orcid.org/0000-0001-5904-5957

## FUNDING

| Funder | Grant(s) | Author(s) |
|---|---|---|
| Ministry of Education and Science of Ukraine | BG-19F, M/51-2024 | Bohdan Ostash |
| Deutsche Forschungsgemeinschaft | Project-ID 439669440 | Mark Helm |
| Federation of European Microbiological Societies | 57507437 | Vasylyna-Marta Tseduliak |

## AUTHOR CONTRIBUTIONS

Vasylyna-Marta Tseduliak, Investigation, Visualization, Writing – original draft | Oksana Koshla, Investigation, Writing – review and editing | Sophie N. Mulartschyk, Data curation, Investigation, Writing – original draft, Writing – review and editing | Virginie Marchand, Investigation, Writing – original draft | Yuri Motorin, Data curation, Supervision, Writing – original draft | Mark Helm, Conceptualization, Funding acquisition, Resources, Supervision, Writing – review and editing | Bohdan Ostash, Conceptualization, Funding acquisition, Supervision, Visualization, Writing – original draft, Writing – review and editing

## DATA AVAILABILITY

All data associated with this manuscript are present either in the main text or Supplemental material. RiboMethSeq sequencing data are deposited to ENA (https://www.ebi.ac.uk/ena/browser/home) under the accession no. PRJEB98478.

## ADDITIONAL FILES

The following material is available online.

### Supplemental Material

**Supplemental material (Spectrum02192-25-s0001.pdf).** Tables S1 to S3; Fig. S1 to S7.

### Open Peer Review

**PEER REVIEW HISTORY (review-history.pdf).** An accounting of the reviewer comments and feedback.

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
