## [Reviewer comments · Microbiology Spectrum]

Microbiology Spectrum

Probing the function of *Streptomyces albidoflavus* J1074 gene *XNR_5296* for SPOUT family ribose methyltransferase

Vasylyna-Marta Tseduliak, Oksana Koshla, Virginie Marchand, Yuri Motorin, Sophie Mulartschyk², Mark Helm, and Bohdan Ostash

Corresponding Author(s): Bohdan Ostash, L'vivs'kij nacional'nij universitet imeni Ivana Franka

Review Timeline:

Submission Date:	July 17, 2025
Editorial Decision:	August 19, 2025
Revision Received:	October 7, 2025
Accepted:	October 29, 2025

Editor: Giordano Rampioni

Reviewer(s): Disclosure of reviewer identity is with reference to reviewer comments included in decision letter(s). The following individuals involved in review of your submission have agreed to reveal their identity: rebecca alexander (Reviewer #2)

Transaction Report:

DOI: <https://doi.org/10.1128/spectrum.02192-25>

Re: Spectrum02192-25 (Probing the function of *Streptomyces albidoflavus* J1074 gene *XNR_5296* for SPOUT family ribose methyltransferase)

Dear Prof. Bohdan Ostash:

Thank you for the privilege of reviewing your work. Your submission has been evaluated by two Reviewers with expertise in the field addressed by your study. It is the consensus of the Reviewers that your manuscript presents both confirmatory findings and novel interesting data. However, both Reviewers noted that, in its current form, some of the key conclusions are not sufficiently supported by the experimental evidence provided. Therefore, substantial revision of the manuscript is required. I would like to invite you to submit a thoroughly revised version of your manuscript that fully addresses all of the Reviewers' constructive comments and suggestions. Please note that the revised manuscript may be subject to a second round of peer review.

I would like to take this opportunity to thank the Reviewers for their excellent work, which resulted in extremely helpful comments to improve the manuscript and strengthen its conclusions.

Please return the revised manuscript within 60 days; if you cannot complete the modification within this time period, please contact me. If you do not wish to modify the manuscript and prefer to submit it to another journal, notify me immediately so that the manuscript may be formally withdrawn from consideration by Spectrum.

Revision Guidelines

Sincerely,
Giordano Rampioni
Editor
Microbiology Spectrum

Reviewer #1 (Comments for the Author):

The manuscript "Probing the function of *Streptomyces albidoflavus* J1074 gene XNR_5296 for SPOUT family 1 ribose methyltransferase" addresses a topic of high interest for the scientific community, namely the role of a putative RNA methyltransferase in uridine methylation. While the subject is relevant, several major concerns arose from my reading of the manuscript.

Abstract

The abstract contains valuable and detailed information on *Streptomyces albus* J1074 and tRNA modifications; however, in its current form, it is difficult for readers not already deeply familiar with the subject to follow. The authors use highly specific terms (e.g., "ms2io6A", "s2Um", "(c)mnm5(s)2U(m)") without providing brief explanations of their biological significance or justification for their relevance to the study. I'd suggest that the authors define or briefly explain uncommon abbreviations and modifications and provide a brief introduction explaining why *S. albus* J1074 and tRNA modifications are important for specialized metabolite production. The authors mixed background information with findings, making it harder for the reader to distinguish what is already known from what is newly discovered in this study. I'd suggest that the authors separate the background, aims, key results, and conclusion into clearly distinct segments.

Importance

This section effectively conveys the significance of studying post-transcriptional modifications in *Streptomyces* tRNAs and highlights the novelty of the work. It is better structured than the abstract, but would still benefit from defining uncommon abbreviations upon first mention.

Introduction

The text successfully combines background on *Streptomyces* biology, bldA-mediated translational control, and tRNA post-transcriptional modifications (PTTMs), while introducing the focus on the gene XNR_5296. The authors should define abbreviations like "Um," "s2Um," "ms2io6A37," and "(c)mnm5U" on first use, even if defined in a figure legend. Some results (e.g., the knockout phenotype) are embedded within the background, making it hard to distinguish new contributions from existing knowledge. The authors should separate known background from the novel results to make the study's contribution more visible.

Results

In the first paragraph, the authors confirm findings that they have already published elsewhere; this section does not provide novel insights.

In the second paragraph, they describe a bioinformatic analysis of the gene under investigation.

In the third paragraph, they present original data on the construction of a mutant strain lacking the putative RNA methyltransferase and demonstrate its role in uridine methylation.

Although these findings could be of interest, the manuscript suffers from significant shortcomings. The results are not presented clearly, and important experimental controls are missing.

Materials and Methods

Generation and verification of the *S. albidoflavus* recombinant strains.

The number of recombinant strains generated is unclear. The complementation strain is not described and is absent from "Table S2. Plasmids and strains used in this work."

It is not stated whether empty vectors were used as controls to assess the potential impact on the strain phenotype.

tRNA digestion and liquid chromatography-tandem mass spectrometry (LC-MS).

The abbreviation "SAM2" should be explained in the text.

The rationale for using *E. coli* ¹²C RNA should be clarified.

Phenotypic analysis of *S. albidoflavus*.

Full genus names should be given at first mention for *B. cereus* and *D. hansenii*.

Beyond the missing control strains, the authors should clarify how they exclude effects of the mutation on other biosynthetic gene clusters (BGCs), as the agar plug antibiotic diffusion assay cannot replace chemical analysis of spent media.

Reviewer #2 (Comments for the Author):

In this manuscript the authors seek to identify the post-transcriptional tRNA modification (PTTM) produced by the XNR_5296 gene of *S. albidoflavus* J1074. They use bioinformatic analysis to demonstrate that the gene sequence appears to be distinct from known SPOUT RNA methyltransferases TrmL, TrmH, and TrmJ. Knockout of the gene suggests that XNR_5296 is responsible for s2Um, likely on tRNA^{Leu}(UAA).

The work is an extension of an earlier manuscript (Koshla J. Bacteriology 2023) and the authors added two more PTTMs to the list for *C. albidoflavus* (k2C and ac4C). But the primary conclusion is that Xnr_5296 has an unusual domain assembly and is not a canonical TrmL, TrmH, or TrmJ ortholog. Knockout of xnr_5296 also leads them to conclude that Um is that product of XNR_5296 and that S2Um is the product of a separate RNA methyltransferase.

The evidence that Xnr_5296 produces Um comes from the loss of LC-MS signal (Figure 4) on the gene knockout. To this reviewer's eye, the signal is decreased but not completely absent. It seems over-reach to say that Um is absent. Also the sentence describing this result (line 230) is confusing: "Notably, both J1074 and D5296 accumulated significant quantities of two nucleoside fractions, presumably ribose-methylated uridines according to neutral loss scan data (m/z 259 Da), whose retention time (8 and 10 min) had no described precedents."

If the N-terminal domain of Xnr_5296 is homologous to eL30, what might be the function of this novel domain? It would be good to have the authors speculate on what role this appended domain might serve. Is it a non-specific RNA-binding domain?

In a few parts of the text, statements seem overly obvious: for example line 79 with reference to *Streptomyces* "In their GC-rich genomes, UAA codon is the rarest one, and its decoding appears to be strictly dependent on cognate tRNA." Certainly decoding is always dependent on a the presence of a cognate tRNA - although perhaps the authors mean that wobble decoding isn't available for this codon? Also at line 87 "The level of charged bldA tRNA is thought to determine the rate of translation of UUA-containing transcripts." Isn't that always true?

Dear colleagues,

Thank you for the comments on our manuscript. We have significantly revised the latter to address the concerns raised in the review. The point-by-point response is given below, here I summarize the major changes to our work. First, we have come to agree that some of the results are written in an unclear manner or are not convincing enough. A better structured text is now provided to highlight novel findings and methodological details. We also carried out additional experiment, RiboMethSeq, to determine the tRNA species affected upon *XNR_5296* knockout. This information is provided a s new section of the results and a new Figure 5. Two new co-authors are listed to the manuscript who carried out RiboMethSeq experiment. As suggested by the reviewers, we re-wrote/corrected some parts of the Introduction and Discussion for more clarity. The revised lines and sections referred to as in this response correspond to lines in the marked-up manuscript file.

Reviewer #1 (Comments for the Author):

The manuscript "Probing the function of *Streptomyces albidoflavus* J1074 gene *XNR_5296* for SPOUT family 1 ribose methyltransferase" addresses a topic of high interest for the scientific community, namely the role of a putative RNA methyltransferase in uridine methylation. While the subject is relevant, several major concerns arose from my reading of the manuscript.

Abstract

The abstract contains valuable and detailed information on *Streptomyces albus* J1074 and tRNA modifications; however, in its current form, it is difficult for readers not already deeply familiar with the subject to follow. The authors use highly specific terms (e.g., "ms2io6A", "s2Um", "(c)mn5(s)2U(m)") without providing brief explanations of their biological significance or justification for their relevance to the study. I'd suggest that the authors define or briefly explain uncommon abbreviations and modifications and provide a brief introduction explaining why *S. albus* J1074 and tRNA modifications are important for specialized metabolite production. The authors mixed background information with findings, making it harder for the reader to distinguish what is already known from what is newly discovered in this study. I'd suggest that the authors separate the background, aims, key results, and conclusion into clearly distinct segments.

*We appreciate the comment. Below we describe how we revised the manuscript to improve its clarity. We defined the abbreviations on lines 34, 37-39, 42 to make it easier for readers. We also revised parts of the background to emphasize the link between *bldA* and antibiotic biosynthesis (lines 27-32), and we outlined the conclusion more clearly due to additional information obtained from the RiboMethSeq data (lines 40-47).*

We also added a new subsection on RiboMethSeq profiling in the Results section (lines 260-287), Methods section (439-469) and Discussion section (306-315).

Importance

This section effectively conveys the significance of studying post-transcriptional modifications in *Streptomyces* tRNAs and highlights the novelty of the work. It is better structured than the

abstract, but would still benefit from defining uncommon abbreviations upon first mention.

We defined the abbreviations (lines 59-60) and revised the results in this section to align with our updated findings (lines 62-64).

Introduction

The text successfully combines background on *Streptomyces* biology, bldA-mediated translational control, and tRNA post-transcriptional modifications (PTTMs), while introducing the focus on the gene XNR_5296. The authors should define abbreviations like "Um," "s2Um," "ms2io6A37," and "(c)mnm5U" on first use, even if defined in a figure legend. Some results (e.g., the knockout phenotype) are embedded within the background, making it hard to distinguish new contributions from existing knowledge. The authors should separate known background from the novel results to make the study's contribution more visible.

*We defined the abbreviations both in the figure legend (lines 111-113, 116) and in the main text (lines 132, 141-142). We also restructured the section by splitting its end to create a new paragraph to outline the results (line 127). In this paragraph, we updated the tRNA species that are substrates for TrmJ in *S. albidoflavus* and *E. coli* (144-147).*

Results

In the first paragraph, the authors confirm findings that they have already published elsewhere; this section does not provide novel insights. In the second paragraph, they describe a bioinformatic analysis of the gene under investigation. In the third paragraph, they present original data on the construction of a mutant strain lacking the putative RNA methyltransferase and demonstrate its role in uridine methylation. Although these findings could be of interest, the manuscript suffers from significant shortcomings. The results are not presented clearly, and **important experimental controls are missing.**

*We thank for the comment. In "An updated description of modified nucleosides produced by *Streptomyces albidoflavus* J1074" subsection, we not only provide a comparison of the overall PTTMs landscape in this and previous work, but also propose the identification of two new PTTMs, k2C and ac4C, along with a suggestion for the absence of (c)mnm5s2Um. To emphasize these new results, we have divided the subsection into three paragraphs (line 167). We provide an additional control for s2Um modification and verified its presence, this time in the presence of authentic standard (see ESM Fig. S1 and lines 168-170).*

Materials and Methods

Generation and verification of the *S. albidoflavus* recombinant strains. The number of recombinant strains generated is unclear. The complementation strain is not described and is absent from "Table S2. Plasmids and strains used in this work." It is not stated whether empty vectors were used as controls to assess the potential impact on the strain phenotype.

We added a missing description of the complementation strain into ESM (Table S2). The wild type strain shown in Fig, 4 is in fact SAM2 harboring empty vector pTES. We apologize for this negligence and now provide correct description in the legend. This vector has been used routinely for Streptomyces experiments and the vector itself affects neither morphology nor antibiotic biosynthesis, at least under our experimental conditions.

tRNA digestion and liquid chromatography-tandem mass spectrometry (LC-MS).

The abbreviation "SAM2" should be explained in the text. The rationale for using *E. coli* ¹²C RNA should be clarified.

The abbreviation SAM2 stands for J1074 derivative strain used through the work, it is explained in Methods (first section) and ESM. The rationale for using E. coli ¹²C RNA clarified in the line 401.

Phenotypic analysis of *S. albidoflavus*.

Full genus names should be given at first mention for *B. cereus* and *D. hansenii*. Beyond the missing control strains, the authors should clarify how they exclude effects of the mutation on other biosynthetic gene clusters (BGCs), as the agar plug antibiotic diffusion assay cannot replace chemical analysis of spent media.

The full genus names of B. cereus and D. hansenii were given at their first mention, at lines 244–245. We provide information on the complementation plasmid (see previous reviewers' comments) and the wild type harboring empty vector, that was used for comparison of morphology.

*Our goal was to screen the overall antibiotic activity and the agar plug diffusion assay fits this goal. This assay provides initial qualitative readout as to whether certain mutation has major (significant – see L. 237) impact on production of specialized metabolites. We indeed are not able to rule out the impact of the mutation on biosynthetic gene clusters whose products are not active against our bacterial test cultures, or that the mutations effects are subtle enough to go unnoticed within our experimental setup. It is reasonable to suppose that these undetected effects are minor, as compared to what we see for *bldA* and *miaA* mutations in our previous works. We therefore did not delve into a deeper analysis of specialized metabolism of the mutant as it was not the objective of our study.*

Reviewer #2:

In this manuscript the authors seek to identify the post-transcriptional tRNA modification (PTTM) produced by the XNR_5296 gene of *S. albidoflavus* J1074. They use bioinformatic analysis to demonstrate that the gene sequence appears to be distinct from known SPOUT RNA methyltransferases TrmL, TrmH, and TrmJ. Knockout of the gene suggests that XNR_5296 is responsible for s2Um, likely on tRNA^{Leu}(UAA).

The work is an extension of an earlier manuscript (Koshla J. Bacteriology 2023) and the authors added two more PTTMs to the list for *S. albidoflavus* (k2C and ac4C). But the primary conclusion is that Xnr_5296 has an unusual domain assembly and is not a canonical TrmL, TrmH, or TrmJ ortholog. Knockout of xnr_5296 also leads them to conclude that Um is that product of XNR_5296 and that S2Um is the product of a separate RNA methyltransferase.

The evidence that Xnr_5296 produces Um comes from the loss of LC-MS signal (Figure 4) on the gene knockout. To this reviewer's eye, the signal is decreased but not completely absent. It seems over-reach to say that Um is absent. Also the sentence describing this result (line 230) is confusing: "Notably, both J1074 and D5296 accumulated significant quantities of two nucleoside fractions, presumably ribose-methylated uridines according to neutral loss scan data (m/z 259 Da), whose retention time (8 and 10 min) had no described precedents."

We thank for the comment and for pointing out the inaccuracies. After revising our findings, we agree with the reviewer that the signal decreases but is not completely absent. This could be due to the presence of Um in different positions in tRNA and Um from rRNA.

As can be seen in Figures 4A and 4B, there are small peaks for Um species for both J1074 and D5296. A closer look at the mass spectra revealed a small difference in the m/z of the molecular ions: 259 and 258.9 Da for SAM2 and D5296 respectively (see the figure 1 below). However, the abundance of the major ion was relatively lower for the D5296 strain. Due to measurement error, it could be interpreted that 258.9 Da is the same ion as 259 Da, thus it is Um. Our subsequent RiboMeth-seq experiment revealed the absence of Um at position 32 in the leucyl tRNA GAG. However, the analysis also revealed the presence of Um at other positions in various tRNAs

Therefore, in agreement with the reviewer's comment, we modified the interpretation of our results (lines 40-41, 249, 254, 303-304) and Fig. 4 ("no signal" to "reduced signal").

Fig.1. Mass spectra of the peaks at retention time 10.8 min. SAM2 is shown in black, D5296 in red.

Regarding the sentence describing the results: we corrected the m/z (259 \rightarrow 258 Da) and modified the sentence to avoid the confusion (lines 250-252). The two massive peaks with RT 8 and 10 min represent protonated form (259 Da) of the unknown nucleoside with mass of 258 Da. Since we are using an extracted ion chromatogram for an ion with a mass of 259 Da, we can see all ions with a matching mass. However, our ion of interest can be distinguished by its respective retention time.

If the N-terminal domain of Xnr_5296 is homologous to eL30, what might be the function of this novel domain? It would be good to have the authors speculate on what role this appended domain might serve. Is it a non-specific RNA-binding domain?

We added speculation on the NTD function (lines 220-221) and expanded the TrmJ-specific motif description (lines 218-219).

In a few parts of the text, statements seem overly obvious: for example line 79 with reference to Streptomyces "In their GC-rich genomes, UAA codon is the rarest one, and its decoding appears to be strictly dependent on cognate tRNA." Certainly decoding is always dependent on a the

presence of a cognate tRNA - although perhaps the authors mean that wobble decoding isn't available for this codon? Also at line 87 "The level of charged bldA tRNA is thought to determine the rate of translation of UUA-containing transcripts." Isn't that always true?

Yes, we mean that wobble decoding isn't available for this codon in Streptomyces. As Rokytskyy et. al (2016) (<https://doi.org/10.1186/s40064-016-2683-6>) stated: "There are no tRNA genes in Streptomyces genomes that would allow UUA codon reading (via correct or wobble interactions) in the absence of cognate tRNA^{UAA_{Leu}}". We add word "single" to emphasize this.

Although the level of charged tRNA is of course a factor in translation of any mRNA, it is not that often a limiting one for the overall expression of a gene. The latter is likely determined by the abundance of mRNA, e.g. transcriptional level regulation. Here we would like to deliver to the reader that bldA, due to its rarity, is sort of a bottleneck in the process of expression of UUA-harboring transcripts. We amended this sentence (L. 91) to better articulate our view of this codon as a regulatory device.

Re: Spectrum02192-25R1 (Probing the function of *Streptomyces albidoflavus* J1074 gene *XNR_5296* for SPOUT family ribose methyltransferase)

Dear Prof. Bohdan Ostash:

Your manuscript has been accepted, and I am forwarding it to the ASM production staff for publication. Your paper will first be checked to make sure all elements meet the technical requirements. ASM staff will contact you if anything needs to be revised before copyediting and production can begin. Otherwise, you will be notified when your proofs are ready to be viewed.

Sincerely,
Giordano Rampioni
Editor
Microbiology Spectrum

Reviewer #1 (Comments for the Author):

The manuscript was largely improved after the revision.